# Study of the Molecular Architectures of 2-(4-Chlorophenyl)-5-(pyrrolidin-1-yl)-2*H*-1,2,3-triazole-4-carboxylic Acid Using Their Vibrational Spectra, Quantum Chemical Calculations and Molecular Docking with MMP-2 Receptor

**DOI:** 10.3390/pharmaceutics15122686

**Published:** 2023-11-27

**Authors:** Mauricio Alcolea Palafox, Nataliya P. Belskaya, Irena P. Kostova

**Affiliations:** 1Departamento de Química Física, Facultad de Ciencias Químicas, Universidad Complutense, 28040 Madrid, Spain; 2Department of Technology for Organic Synthesis, Ural Federal University, 19 Mira Str., 620012 Yekaterinburg, Russia; n.p.belskaya@urfu.ru; 3Department of Chemistry, Faculty of Pharmacy, Medical University of Sofia, 2 Dunav Str., 1000 Sofia, Bulgaria

**Keywords:** 1,2,3-triazoles, anti-cancer drugs, vibrational analysis, scaling, IR

## Abstract

1,2,3-triazole skeleton is a valuable building block for the discovery of new promising anticancer agents. In the present work, the molecular structure of the synthesized anticancer drug 2-(4-chlorophenyl)-5-(pyrrolidin-1-yl)-2*H*-1,2,3-triazole-4-carboxylic acid (**1b**) and its anionic form (**2b**) was characterized by means of the B3LYP, M06-2X and MP2 quantum chemical methods, optimizing their monomer, cyclic dimer and stacking forms using the Gaussian16 program package. The molecular structure was found to be slightly out of plane. The good agreement between the IR and Raman bands experimentally observed in the solid state with those calculated theoretically confirms the synthesized structures. All of the bands were accurately assigned according to functional calculations (DFT) in the monomer and dimer forms, together with the polynomic scaling equation procedure (PSE). Therefore, the effect of the substituents on the triazole ring and the effect of the chlorine atom on the molecular structure and on the vibrational spectra were evaluated through comparison with its non-substituted form. Through molecular docking calculations, it was evaluated as to how molecule **1b** interacts with few amino acids of the MMP-2 metalloproteinase receptor, using Sybyl-X 2.0 software. Thus, the relevance of triazole scaffolds in established hydrogen bond-type interactions was demonstrated.

## 1. Introduction

Despite advances in chemotherapy, drug resistance remains a major clinical concern, creating an urgent need to explore new, more effective and safer anticancer drugs. In this task, the 1,2,3-triazole group is one of the most important classes of nitrogen-rich heterocyclic scaffolds. This heterocyclic class of compounds are considered as amide bioisosteres that can form diverse non-covalent interactions, such as Van der Waals forces and hydrogen bonds with various proteins, enzymes, and receptors with high resistance to enzymatic degradation [1,2]. This fact allows the potential use of these compounds in medicinal chemistry [3], and therefore, they are widely synthesized and tested [3,4,5,6,7]. They have received considerable attention in drug discovery for the development of anticancer agents, especially for lung [5] and breast [7] cancers, and as potential antifungal agents [8]. In addition, the 1,2,3-triazole ring possesses low multidrug resistance, low toxicity, high bioavailability, and stability in both acidic and basic conditions. The nitrogen atom in the 1,2,3-triazole ring is responsible for the enzyme–inhibitor interaction [9]. Recent reviews [3,4,5] have shown an overview of these advances.

In a previous manuscript [10], we described the synthesis, NMR and UV-Vis analysis of four new 1,2,3-triazole derivatives, two of which have now been studied in detail, but from the theoretical and experimental IR and Raman spectroscopy points of view. The selected molecules under study were 2-(4-chlorophenyl)-5-(pyrrolidin-1-yl)-2*H*-1,2,3-triazole-4-carboxylic acid (molecule **1b**) and its anionic form (molecule **2b**) (Figure 1). These molecules have a strong-accepting group, the carboxylic (or carboxylate) group, in addition to the chlorine atom that can lead to special specific properties. They also have phenyl and pyrrolidine groups that confer liposolubility to the structure, which facilitates passage across the cell membrane. 

Carboxylic acids are a rare type of 1,2,3-triazole derivatives. However, they have very attractive functionality, and are widespread in biomolecules. Many azoles, including pyrazoles, isoxazoles, imidazoles, and thiazoles, and some flavone analogs containing this group are widely used in medicine and biology [11,12,13,14,15,16]. Previously, we showed the extremely weak acidity of 2*H*-1,2,3-triazole-4-carboxylic acid (pKa = 7.65–8.08 [17]) and proposed that they do not induce the acidity of biological fluids [9], toxicity and the risk of unfavorable effects—in particular, idiosyncratic drug-induced liver injury [18,19] will be minimal in further clinical trials.

Vibrational FTIR spectra of related triazole molecules have been reported [6,20], although not in such a detailed form as presented here, nor using accurate scaling procedures. For this purpose, several dimeric forms were optimized at three DFT levels, achieving good agreement with the experimental spectra in the solid state and obtaining the following objectives: (i) to characterize and confirm the molecular structure of the synthesized compounds; (ii) to establish the effect of the different substituents on the vibrational spectra, particularly on the triazole ring, and how they affect the molecular properties; and (iii) through molecular docking calculations, to identify at the molecular level the binding sites of our compounds to active MMP-2 metalloproteinase receptors on cancer cells, and therefore, to look for new substituents with more interactions.

Our further investigations with these compounds are devoted to testing their cytotoxic effects on different tumor cell lines in accordance with literature data showing their potential, as well as to use them as ligands for the synthesis of new lanthanide complexes according to our previous studies on the activity of lanthanide coordination compounds with similar biologically active ligands. 

## 2. Materials and Methods

### 2.1. Experimental Details

Molecules **1b** and **2b** were obtained by means of the alkaline hydrolysis of 2-(4-chlorophenyl)-5-(pyrrolidin-1-yl)-2*H*-1,2,3-triazole-4-carbonitrile according to previously developed procedures [17,21]. 1,2,3-triazole-containing molecules form a white powder at room temperature. The IR spectra were recorded in powder form at the spectral range of 400–4000 cm^–1^ on a Brüker IFS-66 FTIR spectrometer equipped with a Globar source, a Ge/KBr beam splitter and a TGS detector. For spectrum acquisition, 50 interferograms were collected. The Raman spectrum was recorded in the 50–4000 cm^−1^ range on a Brüker (Billerica, MA, USA) IFS 66 optical bench with a FRA106 Raman module connected to a microcomputer. The sample was mounted in the sample illuminator using an optical mount without sample pre-treatment of any type. A Nd:YAG laser at 1064 nm was used as an excitation source. The laser power was set at 250 mW and the spectrum was recorded over 1000 scans at room temperature. 

### 2.2. Theoretical Calculations

Computations were mainly carried out using density functional theory (DFT) [22]. DFT calculations on different biomolecules provide results which are quantitatively in good agreement with those obtained at the MP2 level for the same compounds [23,24], thus predicting vibrational wavenumbers more effectively than HF and MP2 methods [25]. In the present study, several levels of theory were used for geometry optimizations and computation of the NBO atomic charges: B3LYP/6-31G(d,p), M06-2X/6-31G(d,p), MP2/6-31G(d,p), B3LYP/6-311++G(3df,pd) and M06-2X/Lanl2dz [26]. All of these methods were implemented in the GAUSSIAN-16 program package [27]. The results obtained were viewed through the GaussView 6.0 graphical interface [28]. 

In the calculation of the harmonic IR and Raman vibrational wavenumbers, the B3LYP method was mainly used, which is one of the most cost-effective DFT methods [25,29], and for this reason it is the most widely used [30,31,32,33,34]. This method has been successfully used in many studies of biomolecules [35,36,37] and in the drug design field [38,39]. The M06-2X method was also used to optimize the dimer structures in the stacking form of **1b** and **2b**, because it appears as one of the best options among the meta-generalized gradient functionals for analyzing dispersion-bound systems [40]. However, the accuracy of this method in the calculation of vibrational wavenumbers is slightly lower than B3LYP, although in a previous study [29], we deduced linear scaling equations and polynomic scaling equations that can noticeably improve the calculated wavenumbers. 

All optimized structures have shown only positive harmonic vibrations (local energy minima). The MP2 method was mainly used to ensure the stability of all optimized structures. Relative energies were obtained by adding zero-point vibrational energies (ZPEs) to the total energy. The Raman scattering activities (*S_i_*) calculated using the Gaussian-16 program were converted into relative Raman intensities (*I_i_*) using the well-known relationship derived from the basic theory of Raman scattering [41].

The ^1^H-NMR values of **1b** in the isolated state were determined at the mPW1PW91/6-311+G(2d,p)//B3LYP/6-311+G(2d,p) level using the Gauge Independent Atomic Orbital (GIAO) method [42]. This level appears to provide precise NMR chemical shifts in organic molecules [43,44], and for this reason it was chosen in the present work. Moreover, this level was implemented in the GaussView 6.0 program, automatically providing the δ_calc_ chemical shift values (in ppm). These δ_calc_ were scaled [45] for the isolated state using the following scaling equation:δ_scaled 1H_ = −0.027 + 1.0933 δ_calculated 1H_

The SYBYL-X 2.0 software [46] package installed on Windows 10 workstations was used to prepare the protein and ligands [47]. The structures of all molecules were built using the SKETCH model in Sybyl-X 2.0. Geometry optimization was performed using Steepest Descent with Gasteiger–Marsili charges and a Tripos force field. Then, optimization using the conjugate gradient was performed, followed by optimization using BFGS [48].

#### 2.2.1. Calculation of Interaction Energies

The interaction energies were calculated in the dimer form of the molecules under study. These computations were carried out only at the M06-2X/6-31G(d,p) level, and the values obtained were corrected for basis set superposition error, according to the procedure described by Boys and Bernardi [49], which was as follows for the present study.

The total counterpoise (CP) corrected interaction energy of the *A* and *A*′ molecules in the dimer *AA*′ is defined as
$$\Delta E_{AA^{\prime }}^{CP}=A^{\mathrm{int}}({AA}^{\prime })+E^{\mathrm{def}}({AA}^{\prime })$$ where
$$E^{\mathrm{int}}\;(AA\prime )=E_{AA^{\prime }}^{AA^{\prime }}(AA\prime )-E_{A}^{AA^{\prime }}\left(AA\right)-E_{A^{\prime }}^{AA^{\prime }}(AA\prime )$$

In this equation, $E_{AA^{\prime }}^{AA^{\prime }}$(*AA*′) corresponds to the calculated electronic energy at the optimized geometry of the dimer (*AA*′), and $E_{A}^{AA^{\prime }}\left(AA\right)$ and $E_{A^{\prime }}^{AA^{\prime }}$(*AA*′) represent the electronic energy of each molecule, *A* and *A*′, of the dimer (*AA*′).

The deformation energy is described as:$$E^{\mathrm{def}}(AA\prime )=E_{A}^{def}\left(AA\prime \right)+E_{A^{\prime }}^{def}\left(AA\prime \right)$$
where $E_{A}^{def}\left(AA\prime \right)$ and $E_{A^{\prime }}^{def}\left(AA\prime \right)$ of each monomer in the dimer can be determined by the following equations: $$E_{A}^{def}\left(AA\prime \right)=E_{A}^{A}\left(AA\prime \right)-E_{A}^{A}\left(A\right)$$
and
$$\;E_{A^{\prime }}^{def}\left(AA\prime \right)=E_{A^{\prime }}^{A^{\prime }}\left(AA\prime \right)-E_{A^{\prime }}^{A^{\prime }}\left(A^{\prime }\right)=E_{A^{\prime }}^{A^{\prime }}\left(AA\prime \right)-E_{A}^{A}\left(A\right)$$
where the parentheses indicate whether the computations in the dimer form were performed for the optimized molecular geometry of each monomer form (*A* and *A*′) or that of the dimer (*AA*′), because the optimized form of each monomer is the same, $E_{A}^{A}\left(A\right)$ = $E_{A^{\prime }}^{A^{\prime }}\left(A^{\prime }\right)$. The superscripts denote whether calculations were performed using the monomer or dimer basis set, and the subscripts indicate the molecular system studied.

#### 2.2.2. Scaling the Wavenumbers

The calculated wavenumbers were scaled to adequately reproduce all experimental wavenumber patterns with enough accuracy, which will markedly improve the obtained spectra [29]. The linear scaling equation procedure was explored, using one (LSE) or two equations (TLSE), in this last case for high and low wavenumbers [29]. These procedures represent a compromise between accuracy and simplicity, and therefore they were the main procedures employed to assign the experimental bands. A new procedure called the polynomic scaling equation procedure (PSE) was also used due to its slight improvement in the calculated wavenumbers than LSE and TLSE. These procedures are based [29] on equations calculated in simpler building molecules, such as the benzene molecule used in the present study. The calculated wavenumbers using the theoretical method were represented by ν^cal^, while the scaled values were represented by ν^scal^. Thus, for the LSE procedure, the equation used [29] was as follows:ν^scal^ = 22.1 + 0.9543 · ν^cal^    at B3LYP/6-31G(d,p) level

The equations corresponding to the TLSE procedure were:ν^scal^ = 29.7 + 0.9509 · ν^cal^    at B3LYP/6-31G(d,p) level for the 1000–3700 cm^−1^ range
ν^scal^ = −16.0 + 1.0009 · ν^cal^    at B3LYP/6-31G(d,p) level for the 0–1000 cm^−1^ range

The scaling equations used by the PSE procedure were:ν^scal^ = −4.2 + 0.9909 · ν^cal^ − 0.00000929· (ν^cal^)^2^    at B3LYP/6-31G(d,p) level
ν^scal^ = 6.5 + 0.9694 · ν^cal^ − 0.00000612· (ν^cal^)^2^    at M06-2X/6-31G(d,p) level

The equation corresponding to the M06-2X/6-31G(d,p) level was only used for the dimer forms of **2b** because of the lower accuracy [32] of this M06-2X method than B3LYP in the scaled wavenumbers.

## 3. Results and Discussion

### 3.1. Molecular Geometries in the Monomer Form

Several selected optimized geometrical parameters in the monomer form are collected in Table 1, namely bond lengths, bond angles and dihedral angles calculated at different DFT and MP2 levels. The label of the atoms shown in Figure 1 is in agreement with that reported by Safronov et al. [10]. The final optimized structure of the neutral form (**1b**) is shown in Figure 1a, while the anionic form (**2b**) is included in Figure 1b. The values of several bond lengths of interest are inserted into these figures. In addition, the total energy (*E*) values at all calculated levels, including the ZPE (zero-point vibrational energy) correction and the Gibbs energy (*G*), are shown in each plot.

Rotation of the carboxylic group –COOH leads to two possible conformations in **1b**. The most stable one appears with a torsional angle ∠C_8_-C_9_-C_11_=O_12_ = 16.0° by MP2, and it is shown in Figure 1. Another conformation is about 10 kJ/mol less stable, and therefore, it is included in Appendix A and will not be analyzed in the present work. When comparing the optimized geometric parameters of these molecules, a large change is observed between the neutral form **1b** and its anion **2b**. That is, the loss of the carboxylic hydrogen in –COOH has a great effect on the molecular structure. 

In these molecules, the phenyl ring has a full planar structure at all calculated levels, as can be seen from the side views of these figures. The triazole ring is also almost full planar, with torsional angle values of less than 1° in the neutral form and less than 1.5° in the anionic form. However, the pyrrolidine ring appears out of plane according to *sp*^3^ hybridization of the cyclopyrrol ring carbon atoms. This lack of planarity is more noticeable in the neutral form than in the anionic form. This can be explained by a weaker O_12_···H_18_ intramolecular H-bond in the neutral form than in the anionic form, which facilitates its lower planarity.

The three substituents on the triazole ring appear slightly rotated and out of this ring plane: (i) The phenyl ring is rotated with respect to the triazole ring, with C_5_-C_4_-N_4_-N_10_ torsional angle values being remarkably higher in **1b** (−10.3° by MP2) than in **2b** (−2.3°). (ii) The pyrrolidine ring plane also appears to be rotated and out of the triazole ring plane with values around 7° by MP2. (iii) The carboxylic group appears remarkably rotated related to the triazole ring, especially in **2b** with MP2 values of the C_8_-C_9_-C=O_12_ torsional angle of 29.8°. This value is almost twice that in the neutral form **1b**. This can be explained by the greater flexibility of carboxylic oxygens in the anionic form with longer C=O_12_ bond lengths: 1.267 Å in **2b**, which is greater than the corresponding value of 1.225 Å in **1b**. This greater flexibility leads to a rotation of O_12_ to be closer to H_18_ and therefore to stronger O_12_···H_18_ H-bonds in the anionic **2b** form than in **1b**. The coplanar form with the triazole ring corresponds to a saddle point. 

These features were also confirmed by means of calculations carried out in a previous manuscript on these molecules, in which the predicted ^1^H, ^13^C-NMR spectra and fluorescent values were in good agreement with the experimental ones [10], and with those found in related triazoles [17,20,21,50]. This non-planarity of the structure is expected to be reduced in the solid state due to intermolecular H-bonds and, therefore, to crystal packing forces that will tend to compress the structure. 

The chlorine atom has a very small NBO (natural bond orbital) negative charge (Table 2), which causes a small negative charge to appear on the C_1_ bonded atom. This feature leads to a small redistribution of the phenyl ring charges, in such a way that the charge on C_4_ becomes ca. 0.04*me* more positive in **1b** and **2**b with the chlorine atom than in its non-Cl-substituted form. Although this change seems small with the chlorine atom, it is enough to result in a slight rotation of the phenyl ring plane related to the triazole ring plane, −10.3° by MP2 in **1b**, and only −2.3° in its anionic form **2b**. In addition, the chlorine atom increases the carboxylic group rotation as compared to its non-Cl-substituted form, as well as the non-planarity of the pyrrolidine ring (N_14_-C_15_-C_16_-C_17_ torsional angle). 

The loss of the H_13_ proton in the **2b** anionic form leads to a noticeable effect on the molecular structure. The main differences appear in the carboxylic group values, with a greater negative charge on the oxygen atoms in anionic form, and with longer C=O and C_9_-C_11_ bond lengths, leading to a greater flexibility of this group. This lengthening of the C_9_-C_11_ bond in **2b** also leads to an increment in the double bond character of N_7_=C_8_ and C_9_=N_10_ of the triazole ring, and consequently the C_4_-N_4_, N-N and C_8_-N_14_ bonds are lengthened. This feature modifies the triazole ring bond angles by ca. 2°, and especially its torsional angles with an increment in the non-planarity of **2b** (Table 1). 

In these molecules, the highest negative charge corresponds to the hydroxyl oxygen O_13_. This atom, along with the other oxygen atoms, seems to be the most reactive. The nitrogen atoms N_7_ and N_14_ also have high negative charges and are involved in the biological functions of these molecules. The highest positive charge appears on the C_8_ and C_11_ carbon atoms because they are bonded to large negative atoms.

### 3.2. Molecular Geometries in the Dimer Forms

In the solid state, molecule **1b** is expected to be symmetrically H-bonded in a cyclic dimer form through the –COOH group, as in related molecules [17] with carboxylic groups [21]. Thus, the dimeric structure of **1b** was optimized, as plotted in Figure 2. These predicted optimized dimer forms for the crystal unit cell were confirmed by comparing the carboxylic vibrations of their IR and Raman spectra with those obtained experimentally in the solid-state sample. Both dimer molecules are almost planar and appear H-bonded with the same bond length, 1.646 Å at the B3LYP/6-31G(d,p) level, but slightly shorter by M06-2X and larger with the 6-311++G(3df,pd) basis set. This H-bond value is slightly longer than that reported in the benzoic acid (BA) dimer at the B3LYP/6-31G(d,p) level, 1.616 Å [51]. This is because of the slight decrease in the negative charge on =O_12_ in **1b** as compared to BA. A longer C=O bond (1.240 Å vs. 1.237 Å in BA) and consequently a shorter C_9_-C_11_ bond length (1.468 Å vs. 1.487 Å in BA) were calculated. In the dimer formation of **1b**, a lengthening of the acceptor C=O bond (1.225 Å in monomer vs. 1.240 Å in dimer) and a shortening of the donor C-OH bond (1.356 Å in monomer vs. 1.319 Å in dimer) were observed, as expected. This shortening in the C-OH bond length of the carboxylic group is proportional to the lengthening of the intermolecular H-bond [52].

**2b** anion cannot be found as a dimer form in the solid-state crystal through an O-H···O bond as in **1b**. However, a comparison of the theoretical scaled vibrational spectra of the monomer with the corresponding experimental ones reveals discrepancies in the stretching vibrations of the –COO and NNN modes. Therefore, to reduce these discrepancies, several stacking forms were optimized with only two molecules in the anionic form. In this model, π–π interactions appear, as well as interactions between the –COO and NNN moieties. In this way, we try to obtain a better calculated IR and Raman spectra and, therefore, an accurate assignment of all bands. Figure 3 shows the three stable forms optimized with the M06-2X method [53]. The most stable dimer corresponds to *form I* (Figure 3a) because it presents the strongest interaction, and the vibrational spectra were studied only for this form. *Form I* is confirmed in the optimized structure at the MP2/6-31G(d,p) level included in Appendix A.

*Form I* appears to be stabilized by several weak H-bonds/interactions through the oxygen atoms and the out-of-plane methylene hydrogens of the pyrrolidine ring. An increment in the twist of the –COO group is found to facilitate these H-bonds/interactions. A high planarity appears between both molecules of this dimer form, and because of this, this structure is expected to be in the crystal unit cell of **2b**. *Form II* (Figure 3b) is slightly less stable than form I, and is also stabilized by weak C-H···O H-bonds/interactions between the –COO group and the pyrrolidine ring hydrogens. In addition, a weak C-H···N H-bond/interaction is observed. However, in this dimer form, the planarity between both molecules is remarkably reduced, and therefore, the crystal packing forces are expected to modify this form. Finally, *form III* with a nearly orthogonal arrangement of both molecules (Figure 3c) is unreliable for the solid state. Other dimer forms with several H-bonds/interactions through the nitrogen atoms were attempted to be optimized, but they were not stable. 

The deformation energy *E*^def^ in the **1b** dimer, 18.6 kJ/mol, is slightly lower than that in the most stable *form I* of **2b**, 21.8 kJ/mol. This can be explained by a larger number of atoms in the stacking interaction than in the **1b** dimer. This deformation energy has similar values in forms *I* and *II* of **2b**, but it is noticeably lower in form *III* (14.0 kJ/mol), which is in accordance with the weaker H-bonds in this dimer. The CP-corrected interaction energy Δ*E*^CP^ is −82.8 kJ/mol in **1b**, in agreement with that reported in the cyclic dimer form of benzoic acid, −81.7 kJ/mol [51].

### 3.3. Molecular Properties

Several thermodynamic parameters, rotational constants and dipole moments were also calculated in the global minimum of monomer and dimer forms and included in Table 3. In general, the values computed using B3LYP were close to those computed using M06-2X, with small differences. Rotational constant values were remarkably reduced in the dimer form, around five times lower than those in the monomer form. By contrast, the C_v_ and entropy (S) values were approximately two times higher in the dimer form than in monomer, as is expected. 

Although the calculated values between **1b** and **2b** were similar, the main difference is in the dipole moment, whose value in the anionic form **2b** was remarkably higher than in the neutral form **1b**. This feature is in accordance with that found experimentally, with a high water solubility of the anion (form **2b**). By contrast, the value in the neutral form **1b** is too low and it has low water solubility. In **1b**, the dipole moment value is slightly lower in the dimer form than in its monomer form, but in the anion **2b,** it is remarkably reduced in its dimer form, in accordance with the –COO group arrangement.

### 3.4. Vibrational Analysis

The calculated wavenumbers in the global minimum of molecules **1b** and **2b** at the B3LYP/6-31G(d,p) level are collected in Appendix A, while a summary of the wavenumbers with high IR or Raman intensity, or those characteristics of the molecular structure, is included in Table 4 and Table 5. The scaled wavenumbers provided via two procedures, LSE (linear scaling equation) and TLSE (two linear scaling equation) are listed in the second and third columns, respectively. The relative IR and Raman intensities (%), the experimental values in the IR and Raman spectra, and the main characterization of the vibrations with their % contribution of the different modes to a computed value (Potential Electron Distribution, PEDs) are also included in these tables. Contributions lower than 10% are not included. Relative intensities are obtained by normalizing each calculated value to the intensity of the strongest one.

Scaling of the IR and Raman spectra was mainly carried out using the TLSE and PSE procedures. Since the PSE procedure leads to the best results, with errors in general being lower than 3%, it was selected as the main scaling procedure to be utilized. The scaled wavenumbers are slightly worse using the LSE and TLSE procedures than with PSE, with the LSE procedure being the worst. Thus, all scaled spectra shown in the present manuscript were obtained by means of this PSE procedure and used for discussion.

A comparison of the whole FTIR experimental spectra with those corresponding to the theoretical scaled spectra obtained using the PSE procedure in the monomer form was carried out and is plotted in Appendix A, while the same comparison with the Raman values is shown in Appendix A. For a better analysis of the different experimental and scaled vibrational wavenumbers of these figures, the spectra were divided into three regions, namely 3700–2700 cm^−1^, 1800–1000 cm^−1^, and 1000–0 cm^−1^ (or 1000–600 cm^−1^ by IR). The IR spectra of the two first regions are shown in Figure 4 and Figure 5, while for simplicity, the Raman spectrum is only presented in the 1800–1000 cm^−1^ range (Figure 6) and the IR spectrum in the 1000–600 cm^−1^ region in Figure 7. The other regions are shown in Appendix A. The assignment of the most intense and characteristic IR wavenumbers is included in these figures.

In a general comparison of the IR spectra, the following was observed: (i) A large difference between the spectra of **1b** and **2b** (Appendix A), which was in agreement with a significant change in the calculated geometric parameters between the neutral and anionic forms. (ii) A noticeable correlation between the scaled wavenumbers in the monomer form and the experimental ones, with only few significant differences, which were reduced with the scaled spectra of the dimer or stacking forms. 

It is noted that most modes appear in the expected ranges. Due to this feature and the very small difference between the experimental and scaled values of most fundamentals, the assignments could in general be considered correct. This assignment of the vibrational bands was carried out through a detailed comparison of the experimental with the scaled spectra. This assignment was analyzed under the following sections: (i) the COOH and COO group modes, (ii) the triazole ring modes, (iii) the phenyl ring modes, and (iv) the C-X modes. The discussion was carried out focusing mainly on sections (i) and (iv) because they involved the most reactive groups.

#### 3.4.1. The Carboxylic COOH Group Modes in **1b** Molecule

The characteristic displacement vectors of these modes are similar for the monomer and dimer forms, although for each dimer vibration, two wavenumbers appear, one corresponding to the in-phase mode (Raman active) and another one corresponding to the out-of-phase mode (IR active). Appendix A collects the calculated (scaled) wavenumbers, along with the experimental ones and the main characterization with the %PED, but only a summary is included in Table 5. The theoretical values correspond mainly to the monomer form of **1b** and **2b**, and to a few values in the dimer form but with noticeable differences compared to the monomer form. The experimental IR spectra are characterized by a broad and complex spectral structure around 2850 cm^−1^, which is characteristic of the carboxylic acid association displayed by hydrogen bonds. The normal modes of the COOH and COO groups were analyzed under the following sections: 

*The C=O modes*: The C=O stretching in the monomer form was calculated with very strong intensity in accordance with the experimental observations, and scaled (by means of PSE) at 1749 cm^−1^. However, this value noticeably differs (ca. 60–80 cm^−1^) from that observed experimentally at lower values. This feature is in accordance with a dimer formation in the solid state through the carboxylic –COOH group. Thus, the scaled values in the dimer form of **1b** at 1699 cm^−1^ (IR) and at 1658 cm^−1^ (Raman) were effectively assigned to the experimental bands at 1690.5 cm^−1^ (IR) and at 1644.2 cm^−1^ (Raman), respectively. In the solid state of BA, this mode has been reported [51] at 1693 cm^−1^ (IR) and 1635 cm^−1^ (Raman), in agreement with our results.

The C=O in-plane bending is characterized by a strong contribution of the δ(C-OH) mode, and therefore it is denoted as δ(COOH) in Table 5. It was predicted (scaled) in the monomer form at 699 cm^−1^. However, in the dimer form, it was scaled at 745 cm^−1^ (IR) with medium IR intensity and at 736 cm^−1^ (Raman) with a weak value, in excellent accordance with the experimental IR band at 747.4 cm^−1^ with medium intensity and the Raman band at 739.7 cm^−1^, also with medium intensity. 

The C=O out-of-plane bending was scaled in the monomer at 704 cm^−1^, very close to that determined in the dimer form at 699 and 697 cm^−1^. The very low IR and Raman intensities predicted for this mode make its determination difficult in the experimental spectra. Only the very weak IR band at 704.9 cm^−1^ was assigned to this mode. A large contribution of this mode was also observed in the scaled wavenumber at 772 cm^−1^. 

*The C-O_13_ modes*: The C-O stretching is noticeably coupled with ν(C=O) and δ(O-H) modes, as well as with δ(C-H) modes, and it was mainly characterized as ν_as_(COO) in Table 5. In the monomer form, it was scaled with medium IR intensity at 1112 cm^−1^, in excellent accordance with the experimental bands at 1112.9 cm^−1^ (IR) and 1117.7 cm^−1^ (Raman). In the dimer form, it was predicted with very weak intensity at 1255 and 1254 cm^−1^, and thus it was not detected in the experimental spectra. The IR band in BA at 1347 cm^−1^ was assigned to this mode [51]. This large difference in the experimental wavenumbers between **1b** and BA is due to the longer C-O_13_ bond in **1b** than in BA, in agreement with the shorter O_13_-H bond in **1b**. The ν_s_(COO) mode was scaled in the monomer form at 1091 cm^−1^ and it was closely related to the experimental IR band at 1088.8 cm^−1^ and to the Raman line at 1090.7 cm^−1^. This mode appears to be strongly coupled with the ν(NNN) stretching of the triazole ring, making its identification difficult. In the dimer form, this mode was predicted with very weak intensity, and therefore it was not included in Table 5. 

*The O_13_-H modes*: A broad and weak IR band at 3516.0 cm^−1^ is observed in the experimental IR spectrum of **1b**, which can only be assigned to ν(O-H) in free COOH groups, based on the scaled wavenumber in the monomer form at 3591 cm^−1^ with medium IR and Raman intensity, and to the experimental IR band reported at 3553.3 cm^−1^ in *p*-methoxy-benzoic acid [54] and in other related molecules [55]. These features are also in accordance with those reported in BA [51], where the monomer form was calculated using B3LYP/6-31G(d,p) at 3763 cm^−1^ (scaled by PSE at 3593 cm^−1^), in excellent accordance with our values in **1b**. Also, in the experimental IR spectrum of the BA molecule [56,57], a band at 3567 cm^−1^ has been found with medium intensity, as we observed in the experimental spectrum of **1b,** but at 3516.0 cm^−1^. This lower wavenumber in **1b** than in BA indicates an intermolecular bond O-H···O stronger in **1b** than in BA. In addition, a very weak and broad experimental IR band is detected at 3420.6 cm^−1^ which can only be due to the stretching of O-H groups weakly intermolecular H-bonded in contact with other molecules. It means that free and H-bonded COOH groups appear in the solid state. 

In the dimer form, the calculated ν(O-H) wavenumber in **1b** is observed to be red shifted (scaled) at 3035 cm^−1^ (IR) and at 2943 cm^−1^ (Raman), and it is predicted with the strongest intensity in both IR and Raman spectra. This large shift indicates strong H-bonds in the solid-state crystal through the hydroxyl groups. This highest intensity in the dimer form is in accordance with that calculated in the BA dimer [51], and that reported in carboxylic acids [58], where it is usually identified according to a broad stretching band near 3000 cm^−1^. In our experimental IR spectrum of **1b,** a broad band is observed at 3054.1 cm^−1^, close to our calculations in the dimer form, but with weak–medium intensity. Bands with strong or very strong intensity were not observed around 3000 cm^−1^. This contradiction, together with the weak experimental bands observed at 3516.0 (IR) and 3599.7 (Raman) and the red-shifted ν(C=O) wavenumber in the dimer form, indicates that in the solid state, not all molecules appear in cyclic dimer forms through the COOH group.

The in-plane bending δ(O-H) appears to be strongly coupled with C=O and C-O modes, as well as with other modes, and it is characterized in Table 5 as δ(COOH). In the monomer form, it appears to be scaled at 1238 cm^−1^, in accordance with the experimental IR band observed at 1241.1 cm^−1^ and the Raman line at 1243.0 cm^−1^. In BA, it was calculated at 1218 cm^−1^ (scaled at 1189 cm^−1^), slightly lower than our calculations in **1b**, and experimentally reported at 1169 cm^−1^ (IR) in BA [56,57]. Thus, the higher wavenumber of this mode in **1b** than in BA is in agreement with the stronger H-bond in **1b** than in BA. 

The out-of-plane bending γ(O-H) mode is clearly characterized as an almost pure mode, and it was scaled at 580 cm^−1^, in accordance with the Raman band at 579.6 cm^−1^. In the dimer form, it was scaled at 998 and 942 cm^−1^ with very weak intensity, and therefore, only the very weak Raman line at 932.5 cm^−1^ could be assigned to this mode. However, in BA, it was reported at 628 cm^−1^ (in free O-H) and at 960 cm^−1^ (in O-H bonded) [51,56,57], in accordance with our results, as well as with the band observed at 546 cm^−1^ in *p*-methoxybenzoic acid [51]. 

#### 3.4.2. The Carboxylate COO Group Modes in **2b**

The ν_as_(COO) stretching mode was predicted with very strong IR intensity in the monomer form of **2b** at 1713 cm^−1^, far away from the very strong IR band at 1606.6 cm^−1^. This red shift toward lower wavenumbers in the experimental spectra can be explained by the intermolecular interactions of these molecules through the COO group that lengthened the CO bond. This feature is in accordance with the strong absorption near 1600–1560 cm^−1^ for asymmetric stretching vibrations reported for the carboxylate group (COO^-^) in the IR spectra of solid-state samples [58]. 

Because planar structures cannot be formed in **2b**, stacking forms were optimized at the M06-2X/6-31G(d,p) level, which are stabilized by interactions/H-bonds of this COO group (Figure 3). These interactions in the stacking form shift the scaled wavenumbers of this mode, but they are also far away from the experimental values. This feature indicates that the packing crystal forces in the solid state are stronger than those calculated in our simplified optimized model. A shortening in the stacking crystal distance between planes will increase the interactions of this COO group and lead to a lengthening of their CO bond lengths, and therefore the wavenumbers will be closer to the experimental ones.

The symmetric ν_s_(COO) stretching mode appears strongly coupled with other modes and distributed over many calculated vibrations. The highest contribution in the monomer form was determined at 1307 cm^−1^ with medium-strong IR intensity, in accordance with the IR band observed with medium intensity at 1301.9 cm^−1^. The values in the dimer form slightly differ from those of the monomer, with scaled values distributed at 1375, 1370, 1362 cm^−1^ and particularly at 1353 cm^−1^ with very strong IR intensity. 

However, only the experimental IR band with medium intensity at 1369.4 cm^−1^ and the very weak shoulder at 1348 cm^−1^ appear close to the scaled values and could be assigned to this ν_s_(COO) mode. Our values also seem to be in strong disagreement with the IR absorption near 1420–1400 cm^−1^ reported for this symmetric stretching in solid-state samples [58], because in this region, we could only detect a very weak band at 1417.6 cm^−1^.

#### 3.4.3. The Triazole Ring Modes

*The NNN modes:* The ν_as_(NNN) stretching appears to be strongly coupled with the ν(C4-N) mode as well as with other ring modes. In **1b,** it was scaled at 1386 cm^−1^ with very weak IR intensity, and therefore it was not observed in the experimental spectrum. A similar wavenumber was calculated in its dimer form because of the small effect of the cyclic dimer on the triazole ring. 

In the monomer form of **2b,** this ν_as_(NNN) mode is scaled at 1320 cm^−1^, but due to its very weak predicted IR intensity, it could not be related to an experimental IR band. This mode is red shifted in the stacking form because of the small π–π interaction of the triazole ring with the COO group. Therefore, it appears spread out in the scaled values at 1328, 1106 and 1100 cm^−1^, with very weak IR intensity. Due to this low predicted IR intensity, it could not be related to experimental bands to confirm this stacking form. The ν_s_(NNN) mode appears scaled at 1360 cm^−1^ with weak IR intensity and could be related to the IR band at 1340.5 cm^−1^. However, in the stacking form, this mode shifts and appears at 1425 cm^−1^ and at 1389 cm^−1^, the latter being in excellent accordance with the weak IR band at 1391.6 cm^−1^, which was assigned to this ν_s_(NNN) mode. 

*The C_8_-N_14_ modes:* The stretching mode was predicted with the highest IR intensity and medium Raman activity in both **1b** and **2b** molecules in good agreement with the very strong band observed in the IR spectra and with the medium intensity line in the Raman spectra. It was scaled at 1566 cm^−1^ in **1b** and closely related to the IR band at 1557.4 cm^−1^ and the Raman line at 1563.2 cm^−1^. This ν(C_8_-N_14_) mode was determined in **2b** at similar values although the experimental IR band appears with medium intensity. 

#### 3.4.4. The Phenyl Ring Modes

The assignments for several of the phenyl ring modes are obvious and require no further discussion. Because of the negligible influence of the dimer and stacking forms of our simplified models in the calculated wavenumbers of the phenyl ring, their values were omitted from the discussion. The Varsanyi notation [59] for a 1,4-disubstituted benzene was followed for the ring mode assignments. Aromatic C-H stretching vibrations are generally observed in the 3120–3000 cm^−1^ region, and they are predicted in the molecules under study with almost null IR intensity and very weak Raman intensity, in accordance with those observed experimentally. 

A ^1^H NMR study [10] showed chemical shifts δ_calc_ values, which were correlated with the C-H-stretching wavenumbers. Therefore, the scaled antisymmetric C-H stretching values in the isolated state of **1b** obtained using the PSE scaling procedure were closely related to the scaled δ_calc_ ^1^H NMR values in the isolated state (Figure 8). This good accordance contributes to confirming our assignment. Perhaps, the small discrepancy observed in the C15H_2_ group of the pyrrolidine ring is due to the fact that its stretching mode appears to be strongly coupled with the C16H_2_ and C17H_2_ groups of the ring. 

The experimental Raman antisymmetric stretching wavenumbers were also related to the experimental δ_calc_ ^1^H NMR values (Figure 8b). Because there are not available experimental values in the gas phase, for this relation we use the ^1^H NMR values reported in methanol [10], although a similar relationship can be obtained with the values in DMSO and 1,4-dioxane. With the exception of the C15H_2_ group, in general a good relationship with the experimental values was obtained, which also confirms our assignment. For simplicity, the values of **2b** were not included in these figures.

#### 3.4.5. The C-X Modes

For a *p*-disubstituted benzene, the ν(C-X) stretching vibrations corresponding to the substituent are the normal modes 13 and 7a that correspond to ν(C_4_N) and ν(C-Cℓ) in the molecules under study. C_4_-N stretching was predicted to be strongly coupled with ν(NNN) stretching and with other modes. Using the displacement vector of the phenyl ring atoms, mode **13** was mainly characterized as ν(C_4_-N) stretching. In **1b,** it appears with medium–strong intensity at 1371 cm^−1^, in good accordance with the IR band at 1378.1 cm^−1^ and to the Raman line at 1379.9 cm^−1^. 

The C-Cℓ stretching in **1b** is coupled with phenyl ring modes, and using the displacement vectors of the ring atoms, it was assigned to mode **7a**. It was scaled at 1082 cm^−1^ and related to the strong IR band at 1088.8 cm^−1^ and to the Raman line at 1090.7 cm^−1^. Due to the strong coupling, this ν(C-Cℓ) mode appears with a noticeably higher wavenumber than in chloro-derivatives [31]. 

### 3.5. Molecular Docking Study

1,2,3-triazole derivatives have been reported to have many different pharmacological properties [4]. Among all possibilities, in the present manuscript, we only explored the binding of **1b** to the MMP-2 matrix metalloproteinase, which is considered [60] to be one of the main models for angiogenesis and tumor development. Therefore, we can obtain a first idea of whether molecule **1b** could act as an effective anticancer drug. For this purpose, the crystallographic structure of MMP-2 was downloaded from Protein Data Bank, PDB ID:1CK7 [60]. The interactions observed between the most stable conformation of our molecule **1b** and different amino acids are shown in Figure 9 from two points of view. The final amino acid positions around **1b** were in agreement with what was previously reported for similar molecules [6]. The four amino acid residues that present binding sites with **1b** were as follows: With aspartic acid (**ASP185**): one of the carboxylic oxygens of ASP185 is weakly H-bonded (at 2.347 Å) to one of the phenyl ring hydrogens of **1b**, as well as to the amino hydrogen of LYS44 (at 2.128 Å). Another carboxylic oxygen of this ASP185 is H-bonded (2.007 Å) to the amino hydrogen atom of the peptide bond of glycine (GLY189), which stabilizes the system.With lysine (**LYS44**): one of the ternary amino hydrogens (NH_3_^+^) of LYS44 is weakly H-bonded (1.979 Å) to nitrogen atom N10 of the triazole ring of **1b**, as well as to the carboxylic oxygen of ASP185 (at 2.128 Å), while another of these amino hydrogens is H-bonded (2.012 Å) to the oxygen atom of the peptide bond of valine (VAL41). It is also noted that the amino hydrogen atom of the peptide bond of VAL41 appears to be specially oriented to the phenyl ring center (2.324 Å), which is rich in π electrons, stabilizing the system.With leucine (**LEU49**): the carboxylic hydrogen H_13_ of our molecule **1b** is H-bonded (1.897 Å) to the oxygen atom of the peptide bond of LEU49, while the carbonyl oxygen O_12_ of this group is H-bonded to the amino hydrogen atom of valine (**VAL51**). Both H-bonds fix our molecule **1b** to this amino acid chain, stabilizing the system. For this reason, we include several additional amino acids, from LEU49 to TYR53, in Figure 9b. These H-bonds slightly opened the (O=C=O) angle to 123.7° vs. 121.7° in the isolated state, and notably rotated the carboxylic group with a value of the torsional angle (C_8_-C_9_-C=O_12_) of 25.6° vs. 10.8° in the isolated state. The weak intramolecular H-bond O_12_···H_18_ calculated in the isolated state (Figure 1a) contributes to this feature. It also contributes to planarity and the disappearance of the H-bonds with amino acid residues. The large flexibility of our molecule **1b** facilitates these interactions.

In addition to these binding sites with **1b** plotted in Figure 9, there are H-bonds between the amino acid chain residues and away from the position of **1b**, which were not included in this figure. Additionally, there are π−π and π−alkyl interactions that stabilize the system. Therefore, the following residues are also included in Figure 9:-Tryptophan (**TRP176**): the phenyl ring of our molecule **1b** appears somewhat close to the TRP176 ring, which indicates a weak π−π interaction. In addition, valine (**VAL41**) is located on the other side of the phenyl ring of **1b**, and its hydrophobic chain appears orientated to this phenyl ring with a short distance between them. Therefore, a π−alkyl interaction is expected, which is in accordance with that reported in other 1,2,3-triazole derivatives [6]. These interactions are confirmed by the noticeable rotation of the aryl ring related to the triazole ring, with a torsional angle (C_5_-C_4_-N_4_-N_10_) of −24.8° vs. −1.7° in the isolated state. The large flexibility of the C_4_-N_4_ bond facilitates these interactions, which appear to be responsible for stabilizing TRP176 and VAL41 close to our molecule **1b**.-The alkyl moiety of leucine (**LEU190**) residue appears somewhat close to the pyrrolidine ring of **1b**, while glycine (**GLY189**) and leucine (**LEU191**) do not present interactions with **1b**, but they show H-bonds with ASP185 that stabilize the system. The significant change in the endocyclic torsional angles of the pyrrolidine ring of **1b**, with a (N_14_-C_15_-C_16_-C_17_) value of −30.2° (vs. −22.5° in the isolated state), may be due to this LEU190 residue. This pyrrolidine ring also appears to be noticeably rotated related to the triazole ring. Therefore, the (C_9_-C_8_-N_14_-C_18_) torsional angle has a value of −7.1° vs. 17.2° in the isolated state. The breaking of the weak intramolecular H-bond O_12_···H_18_ calculated in the isolated state may also contribute to this feature.

## 4. Summary and Conclusions

A detailed study of two 1,2,3-triazoles-containing molecules was carried out from the structural, spectroscopic and possible chemotherapeutic activity points of view. The most important findings of this study were as follows:(1)A conformational study of the molecules under study was performed with the help of MP2/6-31G(d,p) and several DFT levels. Via rotation on the C_9_-C_11_ bond, two conformers were found. The energy difference between both is small, at less than 10 kJ/mol.(2)A cyclic dimer formed through the carboxylic –COOH group was optimized in **1b**, while several stacking forms were optimized in **2b** using M06-2X and MP2 methods.(3)To improve the calculated wavenumbers, three DFT levels and two main scaling procedures were used. The PSE procedure leads to the best results, with errors less than 3%. The scaled wavenumbers are slightly worse using the LSE procedure than the TLSE procedure.(4)The FT-IR and FT-Raman spectra in the solid state of the molecules under study were recorded. For the first time, an accurate assignment of all observed IR and Raman vibrational bands was carried out. The scaled wavenumbers in the monomer, dimer and stacking forms were in good accordance with the experimental bands, which confirms our simplified optimized structures for the crystal unit cell of the solid state.(5)A comparison of the scaled and experimental wavenumbers indicates that in the solid state, free and H-bonded COOH groups appear in **1b**, while stacking forms with COO anion are predicted in **2b**.(6)New relationships between the theoretical scaled and experimental ν_as_(C-H) stretching wavenumbers and the scaled and experimental ^1^H NMR chemical shifts were established in **1b**.(7)Molecular docking calculations reveal four H-bonded sites of **1b** with ASP185, LYS44, LEU49 and VAL51 amino acids of the MMP-2 matrix metalloproteinase. The carboxylic group seems to play an important role through its two H-bonds to LEU49 and VAL51 residues, which fix our molecule to the amino acid chain stabilizing the system.

These findings show that 2-aryl-1,2,3-triazole acids **1b** and **2b** may be good candidates as anticancer drugs. In this regard, additional studies are being carried out in our laboratory with other derivatives.

## Figures and Tables

**Figure 1 pharmaceutics-15-02686-f001:**
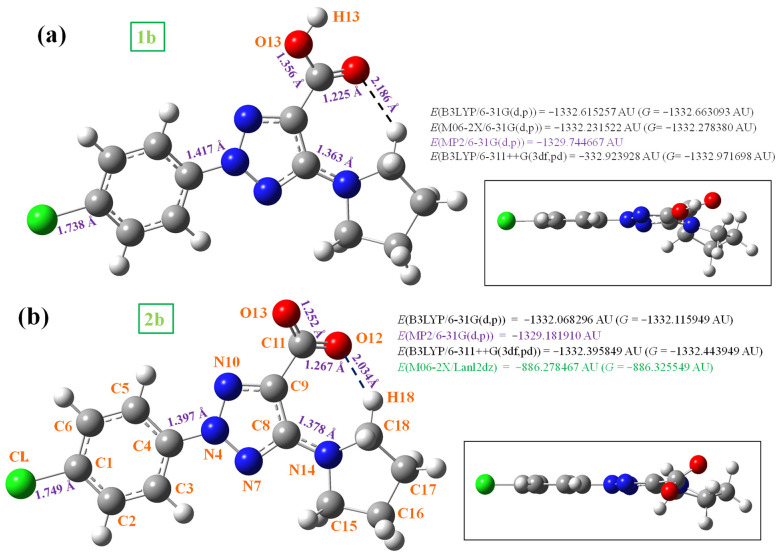
Labeling of the atoms and plot of the structure with front and lateral views forms of (**a**) 2-(4-chlorophenyl)-5-(pyrrolidin-1-yl)-2*H*-1,2,3-triazole-4-carboxylic acid (in short **1b**), where several bond lengths and intramolecular hydrogen bond (H-bond) values of interest are calculated at the MP2/6-31G(d,p) level and included in the figure, as well as the total energy of the system (*E*) including zero-point correction and the Gibbs free energy (*G*); (**b**) 2-(4-chlorophenyl)-5-(pyrrolidin-1-yl)-2*H*-1,2,3-triazole-4-carboxylate anion (in short **2b**).

**Figure 2 pharmaceutics-15-02686-f002:**
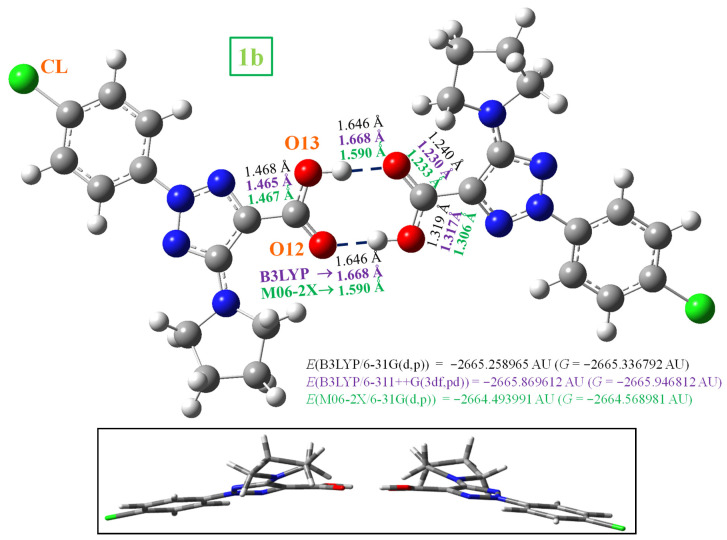
Two views of the optimized dimer form of **1b**. Several bond lengths and H-bond values of interest calculated at the B3LYP/6-31G(d,p) level (in black colour), the B3LYP/6-311++G(3df,pd) level (in violet colour) and the M06-2X/6-31G(d,p) level (in green colour) are also included. 1 AU = 2625.5 kJ/mol.

**Figure 3 pharmaceutics-15-02686-f003:**
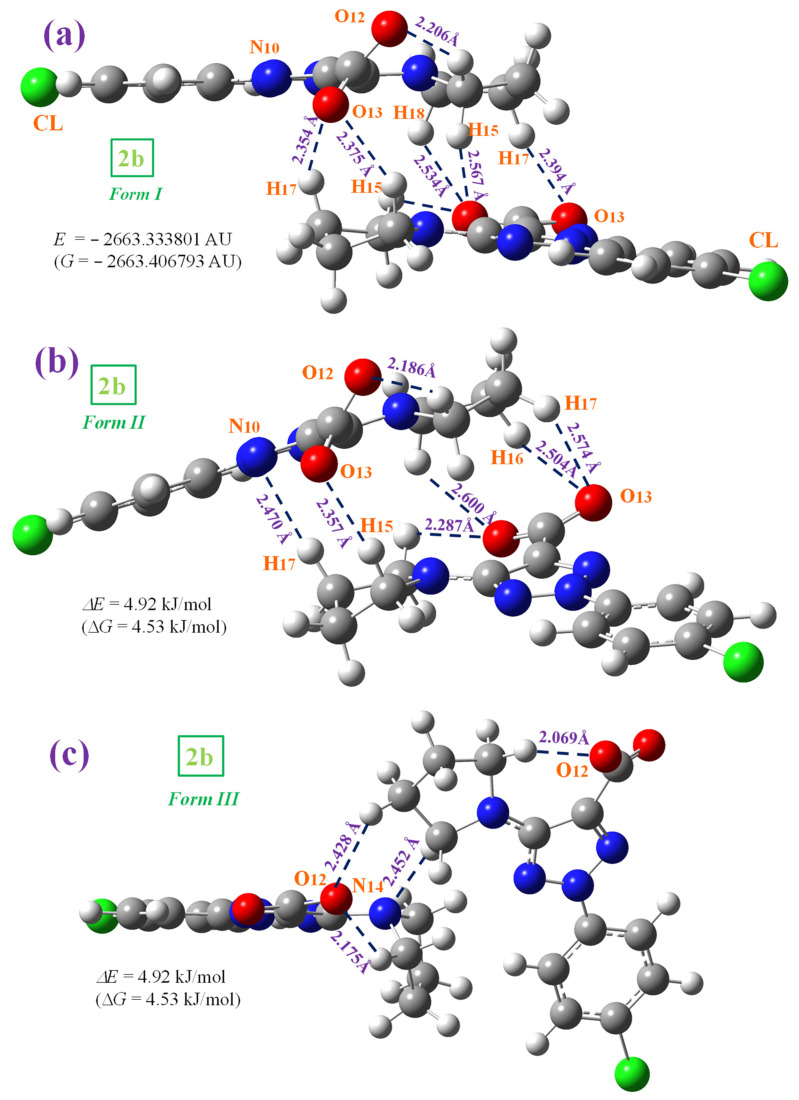
The three optimized dimer forms of **2b**. Several bond lengths and H-bond values of interest calculated at the M06-2X/6-31G(d,p) level are included in the figure. Δ*E* and Δ*G* energy values are related to the most stable form *I*.

**Figure 4 pharmaceutics-15-02686-f004:**
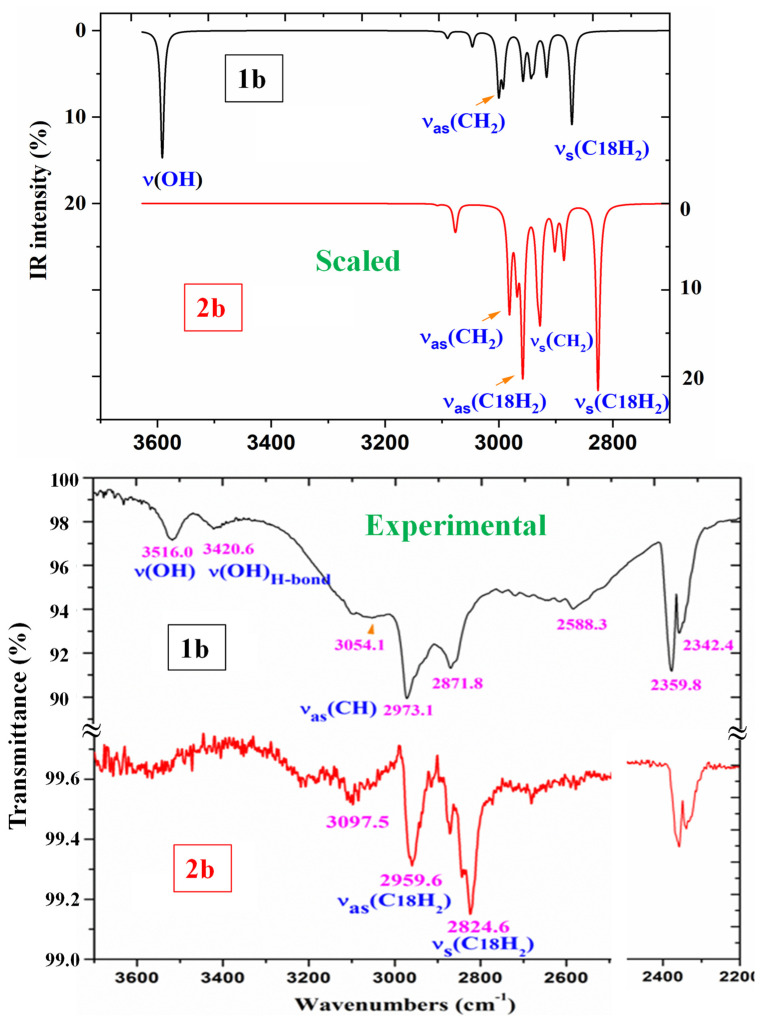
Comparison of the scaled IR spectra of **1b** and **2b** molecules obtained using the PSE procedure in the 3700–2700 cm^−1^ range with the experimental ones in the 3700–2200 cm^−1^ range. The most intense and characteristic vibrations are assigned.

**Figure 5 pharmaceutics-15-02686-f005:**
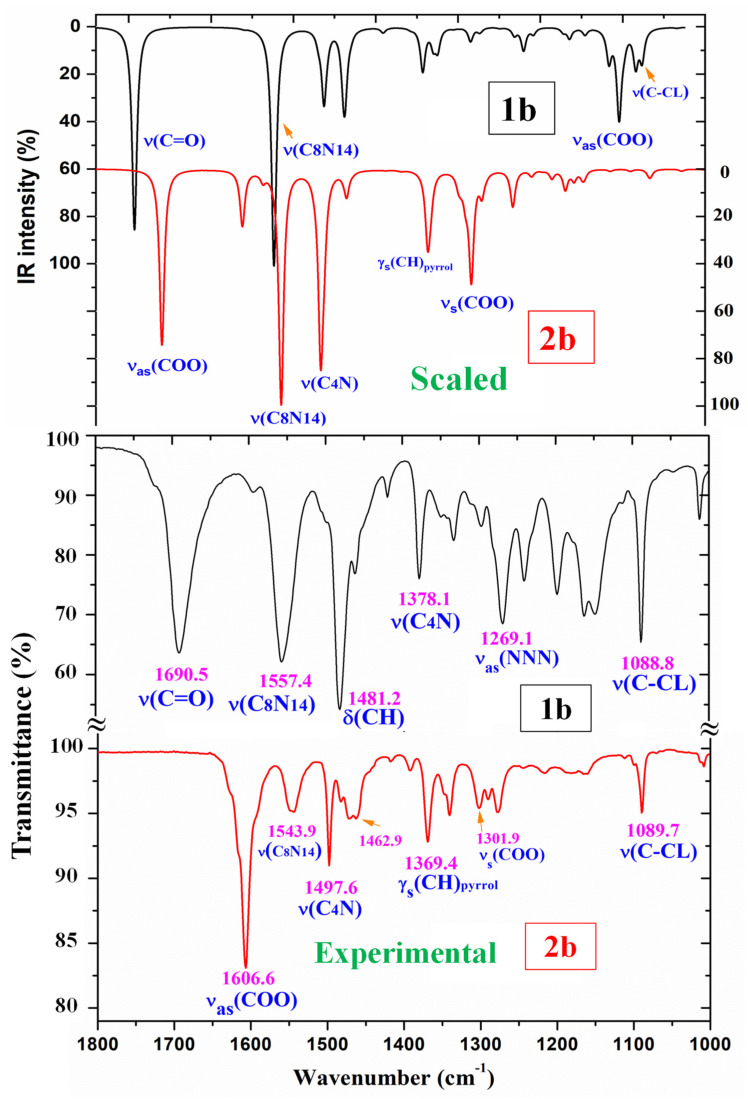
Comparison of the scaled IR spectra of **1b** and **2b** molecules obtained using the PSE procedure with the experimental ones in the 1800–1000 cm^−1^ range.

**Figure 6 pharmaceutics-15-02686-f006:**
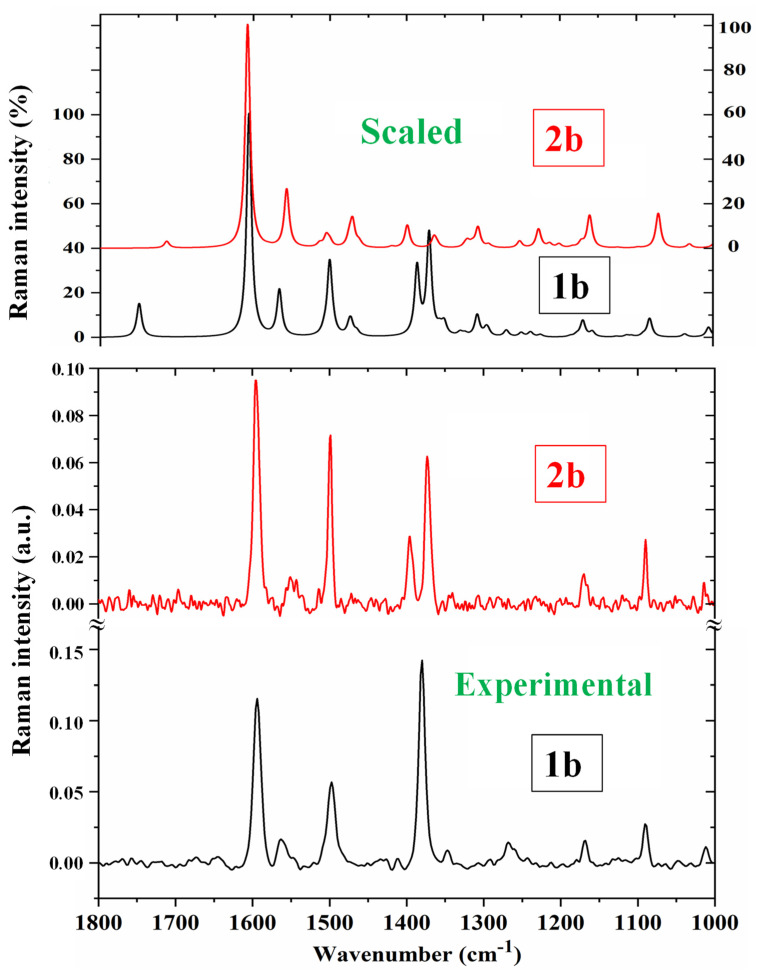
Comparison of the scaled Raman spectra of **1b** and **2b** molecules obtained using the PSE procedure with the experimental ones in the 1800–1000 cm^−1^ range.

**Figure 7 pharmaceutics-15-02686-f007:**
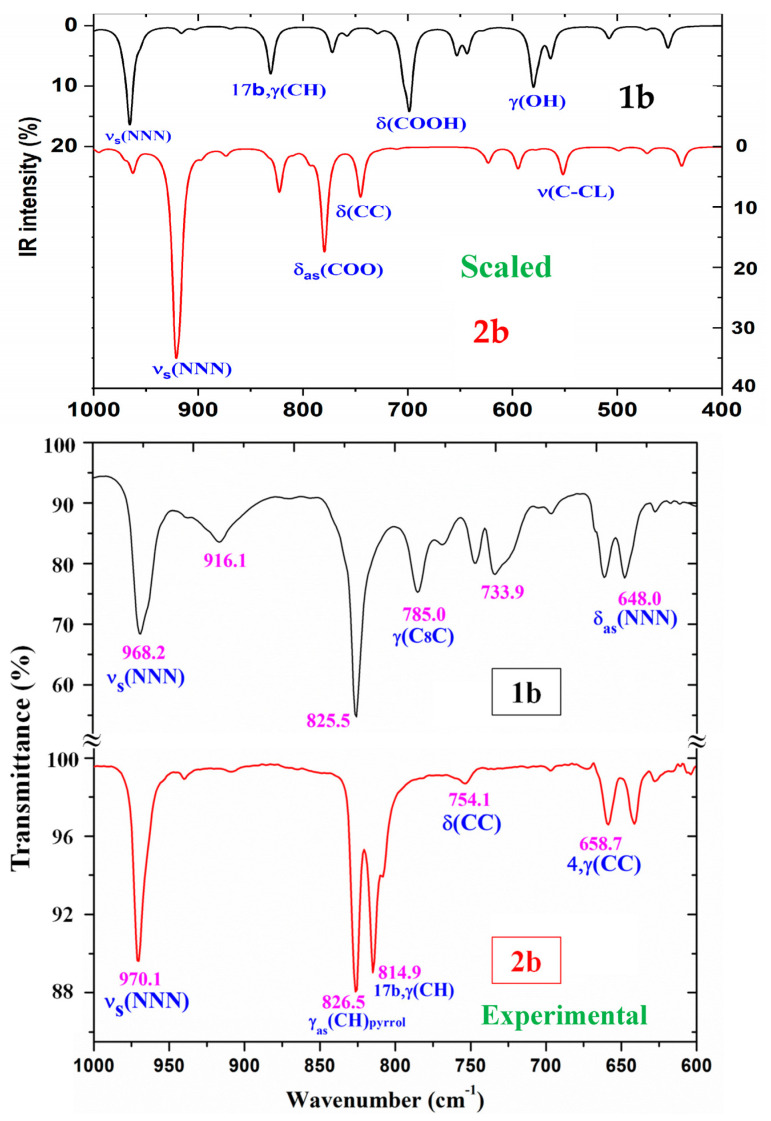
Comparison of the scaled IR spectra of **1b** and **2b** molecules obtained using the PSE procedure in the 1000–400 cm^−1^ range with the experimental ones in the 1000–600 cm^−1^ range. The most intense and characteristic vibrations are assigned.

**Figure 8 pharmaceutics-15-02686-f008:**
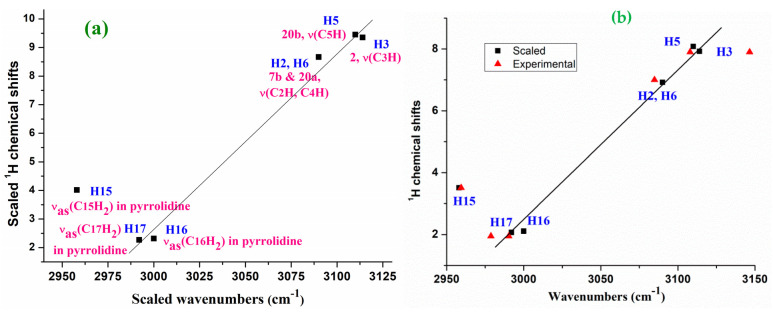
Relationships in **1b** between (**a**) the theoretical calculated ^1^H chemical shifts in the isolated state at the mPW1PW91/6-311+G(2d,p)//B3LYP/6-311+G(2d,p) level and further scaled versus the theoretical scaled stretching ν(C-H) wavenumbers at the B3LYP/6-31G(d,p) level and (**b**) the scaled and experimental ^1^H chemical shifts and the related stretching ν(C-H) wavenumbers.

**Figure 9 pharmaceutics-15-02686-f009:**
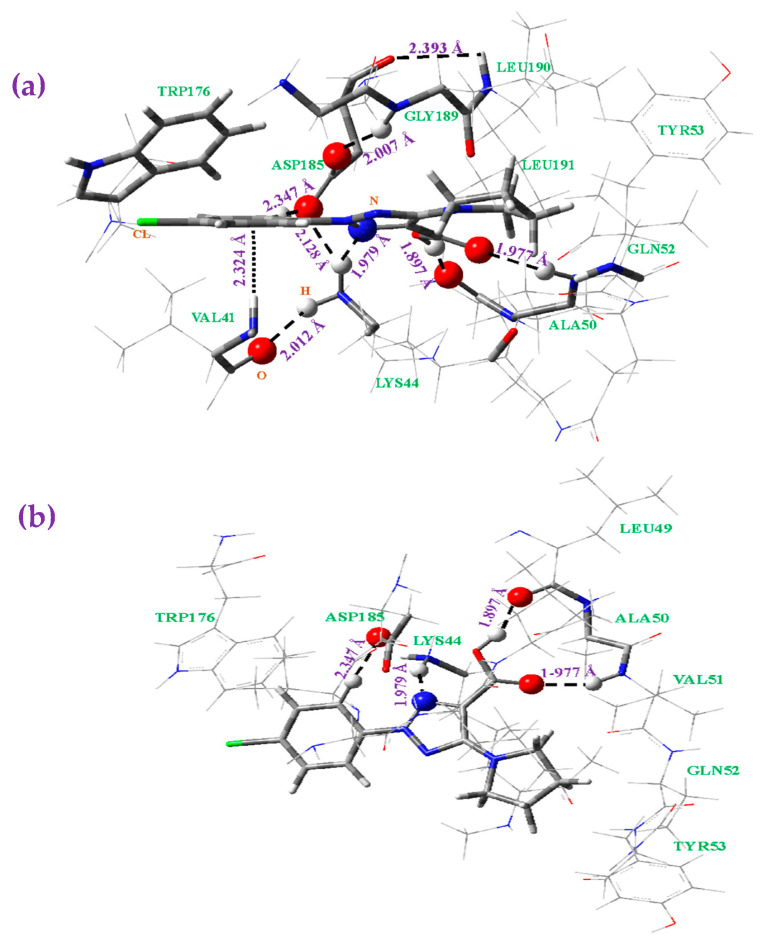
Optimized structure at the B3LYP/6-31G(d,p) level of the system with several amino acids of the MMP-2 receptor forming H-bonds with the **1b** molecule from two viewpoints: (**a**) side and (**b**) front.

**Table 1 pharmaceutics-15-02686-t001:** Several selected optimized geometrical parameters calculated in the monomer form at different levels. Bond lengths (r) are in Å, bond angles and dihedral angles (∠) are in degrees.

Parameters	B3LYP				M06-2X ^c^	MP2 ^a^	
	1b ^a^	1b ^b^	2b ^a^	2b ^b^	2b	1b	2b
r(N_4_-N_7_)	1.350	1.345	1.361	1.355	1.382	1.340	1.350
r(N_4_-N_10_)	1.313	1.306	1.349	1.345	1.365	1.334	1.352
r(C_8_-C_9_)	1.443	1.438	1.453	1.445	1.461	1.423	1.430
r(C_9_-N_10_)	1.344	1.338	1.325	1.316	1.337	1.357	1.348
r(C_9_-C_11_)	1.468	1.465	1.549	1.537	1.535	1.467	1.541
r(CL···O_12_)	10.281	10.224	10.160	9.997	10.253	10.227	10.179
∠(C_4_-N_4_-N_7_)	121.7	121.9	122.4	122.7	122.8	121.5	121.9
∠(C_4_-N_4_-N_10_)	122.4	122.6	122.7	122.9	122.9	121.7	122.0
∠N-N-N)	115.9	115.5	115.0	114.4	114.2	116.8	116.1
∠(N_10_-C_9_-C_11_)	119.5	119.7	121.2	122.7	121.2	119.2	120.7
∠(C_9_-C_8_-N_14_)	133.2	132.9	130.8	130.0	131.4	132.7	130.9
∠(C_9_-C_11_=O_12_)	126.0	125.6	113.4	113.0	113.9	125.6	113.3
∠(C_9_-C_11_-O_13_)	112.3	112.6	115.9	116.5	116.6	111.8	115.4
∠(O=C=O)	121.7	121.8	130.7	130.4	129.5	122.6	131.3
∠(C_5_-C_4_-N_4_-N_10_)	−1.7	−2.5	−0.8	−0.9	−1.3	−10.3	−2.3
∠(N_4_-N_10_-C_9_-C_11_)	175.8	174.3	175.0	176.3	175.7	173.7	175.9
∠(N_10_-N_4_-N_7_-C_8_)	−0.4	−0.5	−1.1	−0.9	−0.4	−0.8	−1.4
∠(N_10_-C_9_-C=O_12_)	−164.9	−161.4	−139.2	−121.5	−140.9	−156.3	−145.6
∠(C_8_-C_9_-C=O_12_)	10.8	12.5	35.2	54.5	34.1	16.0	29.8
∠(C_11_-C_9_-C_8_-N_14_)	4.8	6.5	5.4	3.8	6.9	8.0	3.1
∠(C_9_-C_8_-N_14_-C_18_)	17.2	21.4	14.8	10.7	4.9	24.9	32.0
∠(C_8_-N_14_-C_15_-C_16_)	−163.5	−162.4	−171.6	−168.3	−170.3	−152.0	−162.4
∠(N_14_-C_15_-C_16_-C_17_)	−22.5	−20.7	−18.3	−21.2	−25.1	−23.6	−9.3

With the basis set: ^a^ 6-31G(d,p). ^b^ 6-311++G(3df,pd). ^c^ Lanl2dz.

**Table 2 pharmaceutics-15-02686-t002:** Natural atomic charges (in *me*) calculated in the monomer form at the MP2/6-31G(d,p) level.

Atom	1b	2b	Atom	1b	2b
CL	−0.012	−0.054	N_10_	−0.183	−0.240
C_1_	−0.052	−0.080	C_11_	0.979	0.955
C_4_	0.160	0.210	=O_12_	−0.720	−0.887
N_4_	−0.005	−0.073	O_13_	−0.775	−0.839
N_7_	−0.381	−0.402	N_14_	−0.560	−0.557
C_8_	0.462	0.419	C_18_	−0.211	0.219
C_9_	−0.091	0.031	H_18_	0.263	0.297

**Table 3 pharmaceutics-15-02686-t003:** Molecular properties calculated at the (a) B3LYP/6-31G(d,p) level, (b) M06-2X/6-31G(d,p) level, and (c) B3LYP/6311++G(3df,pd) level.

	Molecular Properties		1b			2b		
			(a)	(b)	(c)	(a)	(b)	(c)
monomer	Rotational constants: (GHz)	A	0.614	0.623	0.619	0.649	0.667 ^a^	0.642
		B	0.138	0.139	0.139	0.136	0.135 ^a^	0.134
		C	0.113	0.115	0.114	0.114	0.114 ^a^	0.112
	C_v_ (cal/mol·K)		64.73	63.88	64.85	63.88	64.01 ^a^	62.84
	S (cal/mol·K)		138.90	136.12	138.77	138.41	139.6 ^a^	136.6
	Dipole moment (Debye)		2.458	2.237	2.414	11.366	11.64 ^a^	12.99
dimer	Rotational constants: (GHz)	A	0.117	0.122	0.114	-	0.206	-
		B	0.023	0.023	0.023	-	0.030	-
		C	0.020	0.020	0.020	-	0.029	-
	C_v_ (cal/mol·K)		133.1	131.2	133.2	-	131.7	-
	S (cal/mol·K)		241.5	234.0	240.1	-	230.2	-
	Dipole moment (Debye)		2.390	2.466	2.719	-	4.070	-

^a^ At the M06-2X/Lanl2dz level.

**Table 4 pharmaceutics-15-02686-t004:** Calculated harmonic wavenumbers (ν^cal^, cm^−1^), relative infrared intensities (A, %) and relative Raman scattering activities (S, %) obtained at the B3LYP/6-31G(d,p) level in **1b** and **2b** molecules.

ν^cal^ 1b	TLSE	PSE	A	S	Experimental ^†^ 1b		Characterization of 1b
	ν^scal^	ν^scal^			IR	Raman	
3123	2999	3000	7	11	2973.1 m	2990.3 vw	ν_as_(C-H) in C16H_2_, C17H_2_ pyrrolidine
1650	1599	1605	0	100		1594.0 vs	8a, ν(C=C) (100)
1609	1560	1566	100	20	1557.4 vs	1563.2 m	ν(C8-N14) (70) + ν_s_(N7CC) (18)
1541	1495	1501	30	32	1500.3 sh	1497.6 s	19a, ν(CC,CH) in aryl (85)
1512	1467	1473	25	6	1481.2 vs		δ(C-H) (40) + ν(C-C11) (35)
1504	1460	1465	2	2	1462.0 m	1451.3 vw	δ_s_(C-H) out-of-phase in pyrrolidine (98)
1422	1382	1386	1	29		1412.7 vw	ν_as_(NNN)(45) + ν(CN)(25) + δ(CH)(20)
1406	1367	1371	18	43	1378.1 m	1379.9 vs	ν(C4N)(35) + ν(NNN)(30) + δ(CH)(25)
1302	1268	1270	0	3	1269.1 m	1268.1 w	ν_as_(NN)(65) + 3, δ(CH) in aryl (25)
1215	1185	1186	2	0	1197.7 m	1196.7 vw	γ_as_(C-H) out-of-phase in pyrrolidine (88)
1198	1169	1170	0	7	1163.0 m	1167.8 m	9a, δ(C-H) in aryl (80)
1186	1157	1158	3	2	1148.5 m	1133.1 w	γ_as_(C-H) in pyrrolidine (81)
1108	1083	1082	13	7	1088.8 s	1090.7 m	ν(C-CL) (60) + 18a, δ(CC) in aryl (40)
1029	1008	1006	2	4	1012.6 m	1011.6 w	12, δ(CC, CH) in aryl (90)
988	973	966	15	10	968.2 m	967.3 s	ν_s_(NNN)(43) + 12,δ(CC)(25) + ν_s_(CN14C) (20)
889	874	869	0	1	--	870.8 m	γ_as_(C-H) in pyrrolidine (98)
849	834	830	8	0	825.5 s	836.1 vw	17b, γ(C-H) in aryl (100)
ν^cal^ **2b**	TLSE	PSE	A	S	Experimental ^†^ **2b**		Characterization of **2b**
	ν^scal^	ν^scal^			IR	Raman	
1652	1601	1607	24	100		1594.9 vs	8a, ν(C=C) (89)
1599	1550	1556	100	26	1543.9 m	1550.6 m	ν(C8-N14) (73) + ν_s_(N7CC) (15)
1545	1499	1505	76	5	1497.6 s	1498.6 vs	ν(CN4) (37) + 19a,ν(CC)(35) + δ(CH)_pyrrolidine_ (15)
1510	1466	1471	10	11	1471.6 m	1471.6 m	ν_s_(CN)in triazol (42) + δ_s_(C-H)in pyrrolidine (35)
1501	1457	1462	0	2	1462.9 m	1466.7 w	δ_s_(C-H) out-of-phase in pyrrolidine (99)
1436	1395	1400	0	10	1391.6 w	1395.4 s	19a,ν(CC,CH) in aryl (40) + ν(C4N)(32)
1400	1361	1365	29	5	1369.4 m	1373.2 vs	γ_s_(C-H) in-phase in pyrrolidine (93)
1395	1356	1360	10	1	1340.5 m	1345.2 w	ν_s_(NNN) (45) + γ_s_(CC,CH) in pyrrolidine (35)
1354	1317	1320	3	2		1318.2 vw	ν_as_(NNN) (36) + γ(CH) in pyrrolidine (30)
1320	1285	1288	1	0	1277.8 m	1280.6 w	3, δ(C-H) in aryl (97)
1258	1226	1228	3	8	1217.0 w	1232.4 w	ν(NN,CN) (68) + γ_as_(CC,CH) in pyrrolidine (19)
1230	1199	1201	4	1		1200.6 vw	ν_as_(NN,CN) (55) + 3, δ(CH) in aryl (22)
1212	1182	1183	9	0	1182.3 vw	1192.9 vw	ν_as_(NN,CN) (48) + ν(CN,CC) in pyrrolidine (35)
1099	1075	1074	2	0		1079.1 vw	ν_s_(NNN) (48) + 18a, δ(C-H) in aryl (42)
1097	1073	1072	2	14	1089.7 m	1089.7 s	ν(C-CL) (32) + 1, δ(CC) (30) + ν_as_(NNN) (28)
1018	998	995	1	4	1008.7 w	1013.5 m	12, δ(CC, CH) in aryl (96)
993	978	971	1	2		970.1 vs	δ(CC,CH) in pyrrolidine (73)
943	928	922	26	1	970.1 vs	923.9 vw	ν_s_(NNN) (60) + ν(C8C)(25) + ν(CC) (15)
852	837	833	1	0	826.5 vs	830.3 vw	γ_as_(C-H) in pyrrolidine (97)
841	826	823	7	0	814.9 vs	821.7 w	17b, γ(C-H) in aryl (99)
564	549	552	5	1		555.5 w	ν(CCL) (38) + δ(NNN)(32) + 6b, δ(CC) (15)
449	433	439	3	0		446.6 vw	δ(CCL) (32) + δ(CC)(25) + δ(NNN) (22)

^†^ Observed frequencies characterized by notation: vs = very strong, s = strong, m = medium, w = weak band, vw = very weak, sh = shoulder. ν_as_: anti-symmetric stretching, ν_s_: symmetric stretching, δ: in-plane bending, γ: out-of-plane bending.

**Table 5 pharmaceutics-15-02686-t005:** Calculated harmonic wavenumbers (**ν^cal^**, cm^−1^), relative infrared intensities (A, %), relative Raman intensities (S, %) together with the scaled values (**ν^scal^**, cm^−1^) in the COOH (of **1b**) and COO (of **2b**) groups. The values in italic type correspond to the dimer form.

Group	Mode	ν^cal^	A	S	TLSE	PSE	IR	Raman	Characterization
					ν^scal^	ν^scal^			
COOH	ν(O-H)	3761	15	17	3606	3591	3516.0 w	3599.7 vvw	ν(O-H) (100)
		*3161*	*100*	*0*	*3035*	*3035*	3054.1 w-m		*Dimer: ν(O-H) out-of-phase*
	δ(O-H)	1269	10	2	1236	1238	1241.1 m	1243.0 vw	δ(OH)(40) + ν(CN)(30) + γ_as_(CH)pyrrolidine (20)
	γ(O-H)	593	9	1	578	580		579.6 m	γ(O-H)(82)
		*964*	*0*	*0*	*949*	*942*		932.5 vw	*Dimer: γ(O-H) out-of-phase*
	ν(C=O)	*1747*	*18*	*0*	*1691*	*1699*	1690.5 vs		*Dimer: ν(C=O) out-of-phase*
		*1705*	*0*	*49*	*1651*	*1658*		1644.2 w	*Dimer: ν(C=O) in-phase*
	ν(C-O)	1139	37	1	1113	1112	1112.9 vw	1117.7 vw	ν_as_(COO)(38) + δ(C-H)(25) + δ(OH) (17)
	δ(C-O)	1117	15	1	1092	1091	1088.8 s	1090.7 m	ν(NNN)(38)+ν_s_(COO)(30)+18a,δ(CH) (25)
		*809*	*0*	*1*	*794*	*791*		799.5 vw	*Dimer: δ_as_(COO) in-phase*
	δ(C=O)	*762*	*1*	*0*	*747*	*745*	747.4 m		*Dimer: δ(COOH) out-of-phase + ν(CC)*
		*752*	*0*	*2*	*737*	*736*		739.7 m	*Dimer: δ(COOH) in-phase*
	γ(C=O)	*714*	*0*	*0*	*699*	*699*	704.9 vw		*Dimer: γ(C=O) out-of-phase*
		*712*	*0*	*0*	*697*	*697*		705.9 w	*Dimer: γ(C=O) in-phase*
COO	ν_as_	1762	75	3	1705	1713	1606.6 vs	1613.3 vw	ν_as_(COO) (99)
	ν_s_	1340	46	8	1304	1307	1301.9 m	1306.7 w	ν_s_(COO)(63) + γ_as_(CH) in pyrrolidine (18)
	δ_as_	797	17	2	782	780	782.1 vvw	767.7 vw	δ_as_(COO) (57) + γ(CC) (32)
	γ_s_	811	2	0	796	793		789.8 vw	γ_s_(COO) (54) + γ_s_(CC9C) (35)

Notation used for experimental bands: vs = very strong, s = strong, m = medium, w = weak band, vw = very weak, vvw = very very weak.

## Data Availability

The data presented in this study are available in this article and Appendix A.

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
