# Peer review of "Study of the Molecular Architectures of 2-(4-Chlorophenyl)-5-(pyrrolidin-1-yl)-2H-1,2,3-triazole-4-carboxylic Acid Using Their Vibrational Spectra, Quantum Chemical Calculations and Molecular Docking with MMP-2 Receptor"

_pharmaceutics, 2023, doi:10.3390/pharmaceutics15122686_

Round 1

Reviewer 1 Report

Comments and Suggestions for Authors

Interesting study with potential for treatment in cancer patients. I have only a few proposals.

Minor points:

1. Please discuss that many drugs for malignancies carry the risk of liver injury, with established diagnosis by causality assessment using RUCAM. Discuss that in the text.

2. Are your chemicals at risk of liver injury, discuss that in the text.

3. Discuss in the text the paper of Jack Uetrecht: Journal of Modern Medicinal Chemistry 2020, 8, 56-64. The question is, whether drugs containing a carboxylic acid functional group are associated with the risk of drug induced liver injury (DILI).

Comments on the Quality of English Language

English is fairly good.

Author Response

I send it in the enclosed attachment

Reviewer 2 Report

Comments and Suggestions for Authors

I can make the following comments on the manuscript: "Study of the Molecular Architectures of 2-(4-chlorophenyl)-5-(pyrrolidin-1-yl)-2H-1,2,3-triazole-4-carboxylic acid as the Potential Anticancer Drug by Their Vibrational Spectra and Quantum Chemical Calculations"

1)      Clearly define the objective of the research in the abstract and at the end of the introduction.

2)      The methodology section of the abstract is poor.

3)      The introduction is short. I recommend creating a new paragraph increasing the details of the previous study [9] and other similar studies in the literature

Comments on the Quality of English Language

Moderate editing

Author Response

I send it in the enclosed attachment.

Reviewer 3 Report

Comments and Suggestions for Authors

Manuscript includes the determination of chemical structure by computational chemistry, and experimental spectroscopy. I guess, this manuscript is better to change a journal to a journal of physical chemistry. The content is much understandable. The following is comments,

1) the authors did not check the effect of anticancer in this manuscript. title. please delete words of potential.....drug.

2) at the end of introduction, chemical structure you want to discuss is better to be moved.

Author Response

I send it in the enclosed attachment.

Round 2

Reviewer 3 Report

Comments and Suggestions for Authors

Manuscript is revised against the comments of reviewer.